# Virtual patients as a tool for training pre-registration pharmacists and increasing their preparedness to practice: A qualitative study

Jessica Thompson●*, Simon White, Stephen Chapman

School of Pharmacy and Bioengineering, Keele University, Keele, Staffordshire, United Kingdom

* j.f.thompson@keele.ac.uk

## Abstract

Virtual patients are an active learning pedagogical tool which simulate clinical scenarios in a three-dimensional environment. Their use in pharmacy education is under-researched in comparison to other healthcare professions. In the United Kingdom, pre-registration training refers to a year of workplace based training which pharmacy graduates must complete prior to professional registration as pharmacists. This study aimed to evaluate pre-registration pharmacists' perceptions on the integration, usefulness and enjoyment of completing virtual patient simulations or non-interactive case studies as part of their training. Pre-registration trainees completed three virtual patient simulations or three non-interactive case studies on the topics of: emergency hormonal contraception, renal function and childhood illnesses. Telephone interviews were conducted with twenty pre-registration pharmacists, exploring their perspectives on the use of the virtual patient or non-interactive case studies. Data was analysed using the five-stage framework approach. Four main themes emerged from the data: case study design; usefulness of the case studies as a training tool; support in pre-registration training; utility of the learning tools. Trainees also identified technical issues they had experienced while completing the virtual patient simulations, specifically with keyword recognition. Pre-registration trainees who used the virtual patients provided comments relating to the novelty, realism and enjoyment in completing them. Trainees in both groups reported developing knowledge and skills from completing the case studies; those who used the virtual patient commented on the development of communication skills and an increase in confidence for practice and those who used the non-interactive cases focused on knowledge acquisition and numeracy. Participants were enthusiastic about virtual patients as a novel training tool which provided an opportunity for learners to practice realistic scenarios in a safe environment. Virtual patients offer the potential to 'bridge the gap' in pharmacist pre-registration sector-related training variation, promote learning through reflection on doing and increase overall preparedness for practice.

**Data Availability Statement:** As per the 'Guidelines for qualitative data', excerpts of the transcripts relevant to the study are within the paper.

**Funding:** The author(s) received no specific funding for this work.

**Competing interests:** The authors have declared that no competing interests exist.

## Introduction

To become a qualified pharmacist in the United Kingdom (UK) requires completion of a four-year undergraduate Master's degree at University and a fifth year of pre-registration training in practice. It is difficult to standardise this pre-registration year and ensure individuals get sufficiently similar experiences to meet the necessary outcomes as set by the General Pharmaceutical Council (GPhC) [1]. A disparity in pre-registration trainee pass marks between the different sectors of training has been identified due to a variety of factors, including the experiences had, the support of the trainees' tutor and the trainees' own resilience and motivation [2]. In order to be 'fit-to-sit' the pre-registration examination, all trainees must demonstrate the 'does' level of competence on Miller's Triangle as per the GPhC [3]. The variation in pre-registration training can affect individual competence development and may have contributed to a theory-practice gap upon qualification [4].

Studies have shown that experience alone may not be enough for individuals to obtain mastery of clinical skills [5]. Simulation-based learning can be instrumental in promoting experiential situated learning and bridging the gap between theory and practice without differences in educational outcomes [6]. Simulation covers a range of learning tools, but those which are higher fidelity are more likely to invoke a greater sense of realism and relation to real-life practice [7]. Virtual patients (VPs) are an active learning pedagogical tool and were utilised in this research. They have been defined as *"a specific type of computer based program that simulates real-life clinical scenarios; learners emulate the roles of health care providers to obtain a history, conduct a physical exam, and make diagnostic and therapeutic decisions"* [8]. A distinguishing factor from other computer-based learning is that VP simulations should unfold in response to learner input [9].

Traditional learning tools encourage individuals to learn in a linear fashion [10] whereas VPs provide the opportunity for users to practice real-world scenarios and make mistakes, promoting a more dynamic learning approach which may result in better retention of knowledge and skills [11]. Providing learners with a safe environment to practice, make mistakes and visualise the consequences of their actions can help encourage reflective learning and prevent errors in the future [12]. Research has illustrated that simulation reduces the pressure and anxiety felt by individuals when completing new tasks to a greater extent than other learning tools, which ultimately may increase users' confidence for completing the same tasks in practice with real patients, and can therefore only be beneficial [13].

The majority of research evaluating VPs has been conducted in medicine or nursing. The first mention of VPs in medical education literature is in 1971 but despite being around for nearly 50 years, their integration into healthcare curricula is limited [14,15]. Multiple designs and technologies exist for VPs, which makes it difficult for researchers to ascertain their effectiveness and adopt 'best practice' [9]. Those classified as higher fidelity have been found to promote development of emotional intelligence, communication skills, clinical reasoning skills, knowledge of a range of conditions and confidence to interact with real patients [7,9,16]. Far less research has been conducted into utilisation of VPs in undergraduate or postgraduate pharmacist education and training, and even fewer studies have evaluated those of a higher fidelity [9,17,18]. Only one study has previously evaluated VP use in pre-registration pharmacist training. This was focused on knowledge and confidence development when delivering a single community pharmacy service [19]. Research evaluating VPs in healthcare education and training has primarily focused on: undergraduate students, a single learning outcome, simulation navigation via pre-defined menu options and rarely have VPs been evaluated with a more-traditional learning tool [20,21]. Whilst this is difficult, as noted by Cook et al (2010), the benefits that VPs can bring to the under-researched area of pharmacy education and

training requires further evaluations to establish their effectiveness as learning tools [20]. The VPs designed for this study utilised both pre-defined menu inputs and free-text inputs to understand the benefits both may offer in the training of healthcare students. This paper reports pre-registration pharmacist perceptions on the integration, usefulness and enjoyment of completing three VP or three non-interactive (NI) case studies as part of pre-registration study.

## Materials and methods

A qualitative approach was adopted for the study. Qualitative data exploring pre-registration pharmacists' perspectives on the use of VP or NI case studies was collected by semi-structured telephone interviews. Telephone interviews were conducted because of the flexibility they offer over face-to-face interviews or focus groups [22].

Ethical approval to undertake the study was obtained from Keele University's Research Ethics Committee.

### Participants

A purposive sample of pre-registration pharmacist trainees completing their training in a UK-based hospital or community pharmacy (training year 2014–15) were recruited to participate in the research by: emails to final year students at Keele University, presentations at regional hospital study days and emails to pre-registration tutors at hospital training sites and one national community pharmacy chain. After trainees had consented to participate, they were randomly stratified into two groups based on their sector of training (community or hospital pharmacy), gender, age and ethnicity to achieve a near-equal distribution; one group received three VP case studies and the other group received three NI case studies. Participants were able to access each case study for one month sequentially, to give a total intervention period of three months. At the end of the three months, participants who had completed all three cases were able to access the alternative type of case studies for one month. This was voluntary and was done to ensure participants had access to both types of learning tool to prevent any disadvantages based on learning style or preference. Trainees were then invited via email to participate in a telephone interview.

### Case study design

Three case studies were created on the topics of: (1) emergency hormonal contraception (EHC), (2) calculation of renal function and (3) childhood illness, to develop a range of knowledge and skills essential to pre-registration training and future practice. Topics were identified from a review of the literature, the aim of the study and conversations between the lead researcher (JT), research team and first year qualified pharmacists, and included areas which trainees identified as being difficult to show their competence in due to training sector variation. The clinical elements of the cases were based on appropriate guidelines and resources which pharmacists use in everyday practice.

### Virtual patient design

The VP simulations were created by the research and digital development teams at Keele University School of Pharmacy and Bioengineering (Figs 1–3) [23].

The VP software has three key parts associated with, and essential to its design: an electronic database containing the VPs responses, a computer generated graphic and a system linking the two together. The electronic database is classed as the 'brain' of the VP which uses

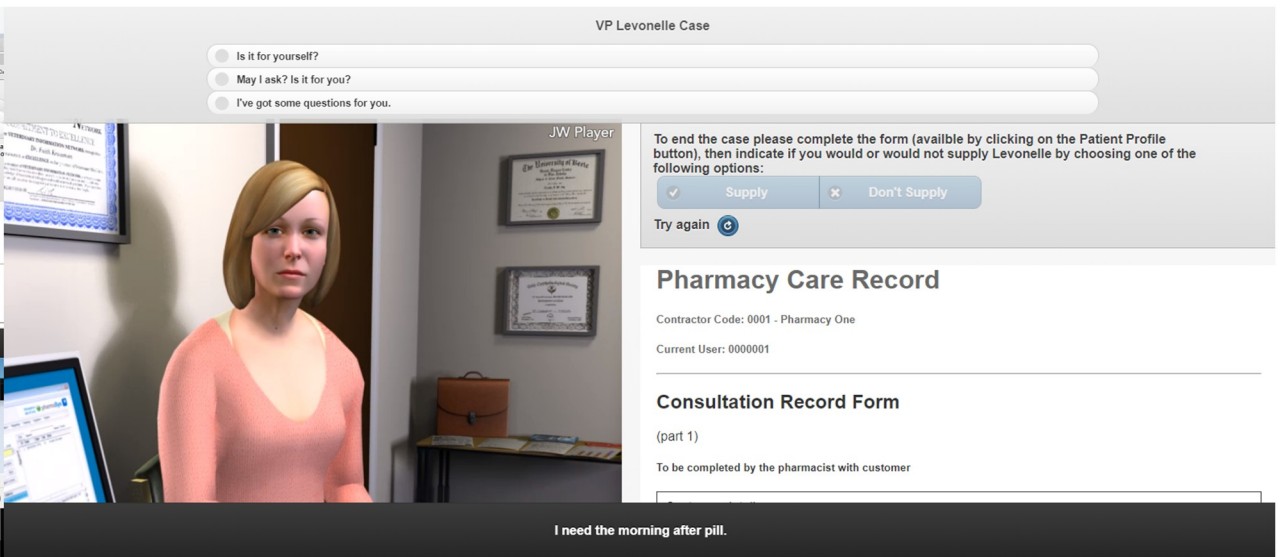

**Fig 1. Screenshot of the EHC VP simulation.**

a decision tree to map the progress of a case and collates the positive and negative feedback to be given at the end of the simulation [24]. The computer-generated graphic is referred to as the 'body' of the VP and is the 3-dimensional (3D) character. The 'heart' is the final part of the system which carries information from the 'brain' to the 'body' allowing a real-time, immediate response.

The script for each case was designed to meet pre-defined learning objectives and ensured equivalent user experiences. Each script was created in consultation with two registered

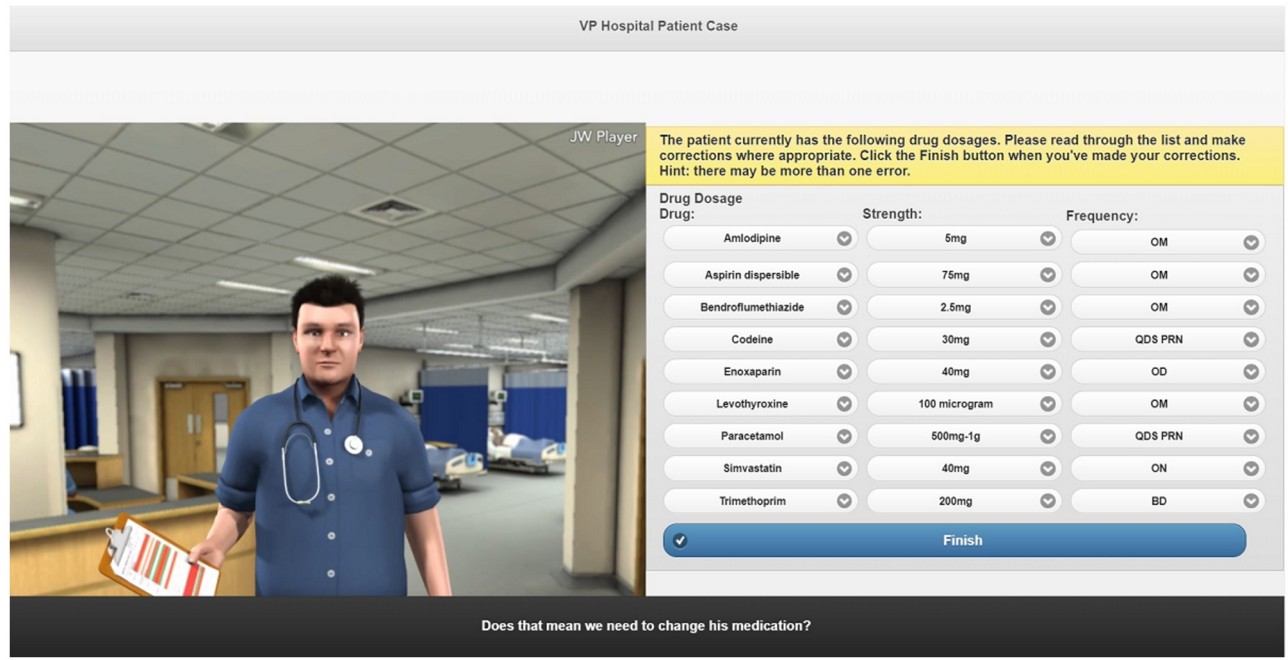

**Fig 2. Screenshot of the renal function VP simulation.**

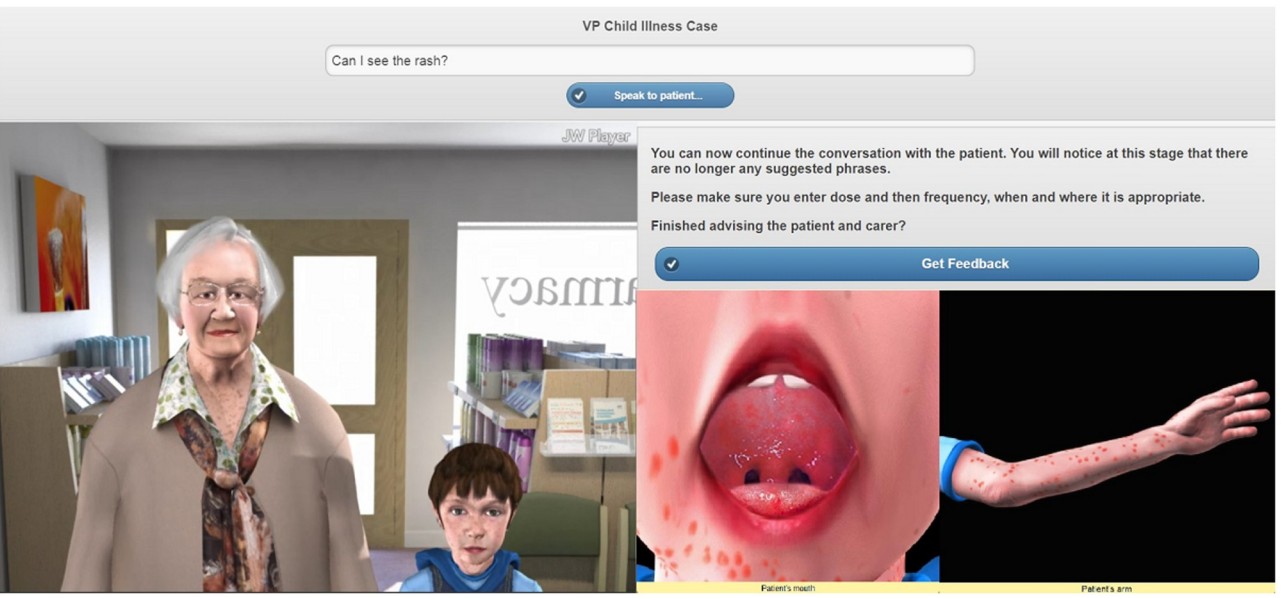

**Fig 3. Screenshot of the childhood illness VP simulation.**

pharmacists who had the relevant expertise. The VPs were designed to express humanistic characteristics through simple body language, movements and pre-recorded voice replies. The VPs used in this research were designed to 'speak' to the user about the decisions they made during the case, and both animated verbal and textual individualised feedback were provided.

Different input styles were used in the VP simulations. The EHC simulation utilised both multiple-choice input and free-text input. This was to help develop trainees' communication skills (i.e. how questions are asked) and knowledge of EHC. A consultation form was integrated into this simulation which guided trainees on the flow of the consultation. The renal function VP simulation integrated hospital notes and a drug chart which individuals needed to use appropriately to complete the case. They were able to interact with the VP via multiple-choice input. The childhood illness VP simulation utilised free-text input and the case was designed such that, when users asked to view the rash an image of the child's arm enlarged, and when asked to check inside the mouth an image of the oral cavity appeared.

The simulations were stored online to allow easy access at a time that suited the trainees. The page layout was defined using HTML with CSS to define the position and colour of screen elements. Characters were modelled, textured and animated using Autodesk's Maya 3D package. Once animated, a series of still images were rendered using the rendering package Mental Ray supplied with Maya. The still images were then composited using Adobe After Effects to create the final .mp4 animation files. For the VP cases using free-text variations, once the 'speak to patient' button was clicked the text was sent to a web service hosted on the same server to process the text. This then returned a code to the site which was processed by the client side JavaScript (using the jQuery library) to determine what animation to play and how the case should behave next. The multiple-choice input cases processed everything client side, skipping the web service step.

**Non-interactive design.** NI case studies were created through consultation with the research team and academic pharmacists at Keele University School of Pharmacy. They existed as a 'Google Form' which trainees were emailed a secure link to. At the beginning of each case study, the intended learning objectives were listed along with an introductory paragraph to

provide background. This was followed by a number of questions and associated text-boxes, in which trainees were required to write their answers before submitting them. Feedback was then provided by displaying the 'model' answer for each of the questions, an explanation of the correct answer and signposting to resources for further reading. Users were required to enter text in each of the textboxes, otherwise they were unable to submit their answers and receive the feedback.

Although the case studies were created on Google Drive, they were still considered to be non-interactive as, apart from answering questions, there was no two-way interactivity between the user and the learning tool.

### Data collection

Initially, 165 participants consented to take part in the study (83 VP group, 82 NI group). Participation decreased with each case study, and 56 trainees were eligible to be invited to a telephone interview. A semi-structured interview guide was developed based on the aims and objectives of the study, literature findings, and the researcher's knowledge of pre-registration training and the VP/NI case studies (S1 File). The interview guide was piloted on an experienced researcher and a newly-qualified pharmacist. As a result, minor amendments were applied to the guide. The broad topics included: perspectives on VPs (or the NI case studies) as a training tool, perspectives on individual support during pre-registration training and improvements to the learning tools. Interviews were conducted by JT until saturation of the broad topics was considered to have occurred by the research team. JT was a new researcher but had experience conducting interviews for previous research, had undertaken training and conducted pilot interviews for this study.

The interviews were audio-recorded to enable full, verbatim transcription. Reflective notes were also made during and after each interview. This included the noting down of any important comments or information which could have been missed in the transcribing process and allowed for reflection on the interview process itself, such as the flow of questions, wording of questions and how responsive participants were to certain questions. Participants were assured of confidentiality, and informed consent for the recording of the interview was obtained.

### Data analysis

The interviews were anonymised (participants were given a number) and transcribed verbatim by an experienced transcriber. The transcripts were then checked for accuracy by JT and amendments made where appropriate. Data was analysed using the five stage framework approach: familiarisation with the data, identifying a thematic framework, indexing, charting, and mapping and interpretation [25]. This approach allowed both a structured data analysis and a complete exploration of the data, to capture previously unidentified ideas and themes. A process of constant comparison was adopted throughout the analysis; interview transcripts were compared with each other to establish analytical categories and identification of similarities or differences of opinions. It was an inclusive process and views which differed from the majority were coded and included in the thematic framework. Themes were discussed within the research team until agreement was reached to improve reliability and quality of the findings. Microsoft Excel was used to facilitate qualitative data management.

## Results

Twenty interviews were conducted; nine trainees in the VP group and eleven trainees in the NI group participated. The remaining 36 participants who were invited for a telephone interview did not respond to the invitation. Despite all having the opportunity, only three trainees

from the NI group and one trainee from the VP group reported that they had used both learning tools at the time of the interview.

In the VP group, five trainees were from the hospital sector and four were from the community sector. In the NI group, six trainees were from the community sector and five were from the hospital sector. Both groups had more female participants (VP group: seven female, two male; NI group: ten female, one male). Interviews lasted on average 30–60 minutes and were conducted at the trainees' convenience, including after working hours and at weekends.

Four main themes emerged from the data, which were consistent with trainees in the VP group and the NI group: case study design; usefulness of the case studies as a training tool; support in pre-registration training; utility of the learning tools. Participants also commented on any technical issues they had experienced while completing the case studies, such as accessibility and usability.

## Case study design

**Realism.** The theme of realism encompassed views on the topics, design and interactivity of the case studies. The majority of trainees who used either type of case study felt the topics were "*really relevant to practicing as a pharmacist*" [Participant 2], and enabled them to put into context what they were learning.

Trainees in the VP group reported that the way the case studies were created with different patients and settings increased their realism. The ability to 'talk' to the patients whilst receiving pre-programmed responses made them feel more realistic.

*". . .you've got to interact with the patients in order to complete the case so they keep you interested and you have to end the consultation like you would do in real practice, instead of just submitting a form with your answers on. It actually made me think about the kinds of things I would say to a patient in real life."*

[P9]

Pre-registration trainees reported that the design of the NI cases allowed them to practice what they were learning.

*". . .it's good because patients aren't going to be textbook patients. . . you can read the topic and try to apply it to a particular patient or patient subgroup. . ."*

[P12]

Suggestions were made to improve the realism of the case studies. The most common improvement that was suggested related to increasing the difficulty or number of problems within a case to reflect a more realistic patient: "*. . .pulling up interactions or making you think in a way that's not just simple [patient medical history taking] questions. . .not just there's the problem, there's the diagnosis and that's the answer, as a lot of patients are more complex*" [P1]. Additionally, utilising more visual aids in the case studies and enabling interactivity with medicines, prescriptions or other items that pharmacists would use daily in a pharmacy were suggested.

**Feedback.** All pre-registration trainees reported that the immediate feedback provided at the end of each case study was useful. Those who completed the VP cases reported that the animated, individualised feedback was a particularly helpful addition to their learning:

*"I think when it [the feedback] was really specific it was the best for me, because then it showed me exactly what I'd missed."*

[P12]

Those who completed the NI cases also reported the feedback as useful but considered it quite general and personalisation would have made it more helpful—such as showing their individual answer next to the 'model' answer as trainees *"could not always remember what [they] had written in the answer box"* [P16].

**Technical issues.** Technical issues were reported for the VP case studies. The main problem associated with the VP technology was the recognition of user inputs when using free-text due to the keyword database not picking up all spellings, words or phrases:

*". . .if the question wasn't in the system then there was no answer from the patient. . . it's quite advanced technology but if you try and have a conversation you're almost limited because there's already a programmed set of questions. . ."*

[P8]

This, however, was not felt to distract from learning by the majority of trainees with one stating *"I don't think it impacted on my learning. . .I still got to see the point that I should have been making in the feedback"* [P6]. Trainees also reported that they used this as an opportunity to think of different ways to phrase questions and ensure they knew the proper spelling of conditions and medicines, in case they needed to interact with a patient in this way in real practice.

A second problem was the accessibility of the VP case studies which some trainees found troublesome with certain web browsers. This was easily resolved by switching to another one but did appear to reduce the remote accessibility on some smart devices.

## Usefulness of the case studies as a training tool

**Experiential learning.** All trainees who completed the VP case studies reported that learning by doing was the most effective way for them to develop their skills and knowledge of being a pharmacist.

*". . .. it's better to do something virtual than just answer questions. . .I am quite happy to read a book and learn but I struggle to just recall it if I don't put it into practice and this let me learn in a more interactive way. . ."*

[P8]

Those trainees who completed the NI case studies reported the ability to apply their learning, however the sense of user responsibility and reflection on learning was not reported.

*"I think that they [the case studies] helped to provide a deeper learning and helped to apply my learning into case study scenarios. . ."*

[P16]

**Development of skills.** When trainees were asked about the skills they had developed from completing the case studies, the most commonly reported were communication and calculation. Over half of the trainees in the VP group reported that the simulations helped

improve their communication skills. This was reiterated by one trainee who had used both case studies, in which they reported that the NI case studies did not promote this development.

*"I think it [communication] ties in more with the virtual cases when you're actually asking the questions. . . but for the paper cases you've just got to type your answers in. I think the virtual cases did help me think about how I was going to phrase my sentence when I speak to a patient. . .so they don't feel anxious and they are comfortable talking with me. . ."*

[P18]

Calculation skill development was reported by trainees who completed the NI case studies but not by those who completed the VP simulations; these trainees' comments focused on the more patient-centred skill development. Calculation skill development was reported as being useful, with participant 10 stating *". . .you can never have too many calculations; it's such a major part of the [GPhC pre-registration] exam*!"

**Development of knowledge.** Trainees in both groups reported that completing the case studies improved their knowledge, identified areas where their knowledge was lacking and encouraged self-directed learning. Most trainees appeared to understand the requirement of a pharmacist being a lifelong learner and were happy to read around the topics and use other resources to improve their knowledge. Some, however, did report that completing the cases should have sufficiently developed their knowledge without having to do further reading.

*"It was more indirect learning because even though I looked up a lot of stuff in the reference sources that I came up with, when the answers came back and I saw the reference sources that were used, it broadened my knowledge a lot. . . I wouldn't have thought to go and look in some of those resources to get the answers."*

[P9]

**Development of self-confidence.** All trainees in the VP group and the majority of trainees in the NI group reported that using the cases improved their confidence for the pre-registration examination. Trainees from the VP group also reported feeling confident for future practice, but this was not reported by any trainees who used the NI cases. The VP simulations were reported as providing a safe environment to practice, develop and apply knowledge and skills before going out into practice as the responsible pharmacist, which made them feel more confident.

*"These virtual patients are a 21ˢᵗ century learning tool and are as close as you can get to real life, to situations you will have to deal with every day when you qualify and are on your own. . . .it gives you confidence because you've done it virtually and if you make a mistake in the simulation you haven't killed anyone and you can try again. . ."*

[P3]

## Support in pre-registration training

**Resources.** Pre-registration trainees commented on the types of resources they would have found useful during their pre-registration training year. Comments related to resources being novel, flexible and reputable. The majority of trainees reported that they had not used any type of VP prior to their involvement in this study, and many comments were received

regarding the novelty of the technology as it appeared to provide a different way of learning which was enjoyed and felt more like a game.

*"I haven't used anything like this before. . .it's up-and-coming technology and I think I got more out of it than with just the normal one [the NI cases]. . ."*

[P6]

Pre-registration trainees also reported that the VP cases gave them a degree of flexibility as they could be completed anywhere as long as they had an internet connection, allowing them to fit the learning into their daily schedules. The NI cases required more reading and writing which was reported as difficult to do on a smaller screen. Both types of case study were reported as being reputable by trainees as they were up-to-date and grounded in a solid evidence-base. It was reported as being *"difficult to find reputable resources. . .there's a lot of textbooks and online questions. . .which can be out-of-date, or past papers, which again are so old now. . ."* and that *"having virtual cases which are easy to access, are up-to-date, are what we need. . .we can be assured that the information we're getting is correct which makes it a much better revision aid"* [P1].

**Pre-registration variation.**    A major reason for trainees reporting the increased need for support appeared to stem from comments around pre-registration training variability, with it being recognised that *". . .everybody has such different experiences and support"* [P3]. Trainees recommended case study topics which they suggested would be beneficial to the training year. These primarily covered a mixture of community-based (minor ailments) and hospital-based (clinical management of long term conditions) topics to allow individuals to practice areas they were unfamiliar with and increase confidence in areas where they may see themselves working in the future. In total 38 topic ideas were received which encompassed the broad range of skills and knowledge within 'pharmacy practice' (S1 Table)

## Utility of the learning tools

**Use in pre-registration training.**    Trainees reported a number of areas where the VP simulations could benefit pre-registration training, particularly for individual revision, group learning, as preparation for an Objective Structured Clinical Examination and as evidence for competency development. Trainees reported that the NI case studies would be useful as revision tools and for competency development. It was noted that the VPs could be used as replacements for role-plays with a tutor or member of staff and would help individuals become confident in a variety of areas they may not have real-world experience in. These comments were not received for the NI case studies.

*"If you role-play with someone or when somebody's watching you talk to a patient you just feel like you're being tested. But with the virtual scenarios, I think you'd be a bit more relaxed and sort of learn from it rather than feeling embarrassed. . ."*

[P5].

**Future use.**    All trainees reported that they would use the learning tools in the future. The most common use was for Continuing Professional Development (CPD) as *"you do the learning and then you do like a patient case scenario on top of that and then that's a really good CPD because you've got the knowledge and you've tried to apply it before you actually go out there to use it in practice"* [P20]. Trainees also commented on using the cases to support changing work sectors and towards diploma work or other postgraduate courses, such as prescribing.

## Discussion

This is the first study to explore pre-registration pharmacist perceptions on the integration, usefulness and enjoyment of VP case studies or non-interactive case studies. There is currently only one other study which evaluated a VP for training pre-registration pharmacists on a single community pharmacy service. This study brings new findings to the current literature due to the free-text input and animated feedback associated with the VPs used, the additional evaluation of a more-traditional learning tool and the number of case studies utilised and their associated learning outcomes.

The findings identified four broad and interrelated themes: case study design; usefulness of the case studies as a training tool; support in pre-registration training; utility of the learning tools. Pre-registration trainees who used the VP simulations provided comments relating to the novelty, realism and enjoyment in completing them. Trainees in both groups reported developing knowledge and skills from completing the case studies; those who used the virtual patient commented on the development of communication skills and having an opportunity to apply their learning and those who used the non-interactive cases focused on knowledge acquisition and numeracy skill development. Trainees who used the VP reported feeling confident for practice which was something not commented on by trainees who used the NI cases.

A key finding of this study is that variation in pre-registration training resulted in all trainees discussing the need for more support and resources to feel confident for both the pre-registration examination and future practice. The interactive nature of the VPs seemed to promote experiential learning, which has previously been identified as essential for healthcare students in order for them to understand how the skills and knowledge they are developing translate to real-world practice [26,27]. The variability of pre-registration training has been well-documented by the GPhC and providing access to a resource which can promote experiential learning of a range of topics may be beneficial at reducing variation and allow for greater standardisation in training experiences [2,4,9]. It has previously been identified that VPs may add little value when learners are at the lower levels of Bloom's taxonomy [28], and they should instead be used when knowledge is combined with skills and applied in problem solving scenarios, or when direct patient contact is not possible [21]. Pre-registration trainees in this study reported the development of real-life complex skills and aspects of learning from using the VP, whereas the simple skills and knowledge reported by using the NI case studies are those which have previously been shown to be developed successfully from 'rote learning' [29].

Effective communication is essential in ensuring positive health outcomes and best patient care. An increased emphasis has been put onto pharmacists' consultation skills, which are likely to become even more important with the advancement of a pharmacists' role in the coming years [4, 30]. Research has found that after undergraduate education, medical graduates' communication abilities may deteriorate as a result of non-continuous training in this area [31]. Pharmacy undergraduates have less clinical placement exposure than other healthcare professions, thus the creation of learning tools which can aid the practice and development of communication skills may be useful, and this may span across all healthcare disciplines. Previous research has found mixed results regarding the development of communication skills from VP use, with fidelity of the tool being a major factor [32–35]. The free-text input in the EHC and childhood illness case studies are fairly unique as the standard navigation through a case is via pre-defined menu inputs [21]. Pre-registration trainees noted that, despite any technical issues, the VP simulations which utilised the free-text input helped them think about how to phrase patient friendly questions, establish a rapport and ultimately provide better patient care; this was not reported by those trainees who used the NI cases.

Both types of case study were an additional resource to participants' pre-registration training year. It is therefore no surprise that trainees reported the development of skills and knowledge from using either type, as they were intended as a learning tool. The main distinction between the two learning tools was the increased realism of the VP, which was primarily due to the design and interactivity of the software. The VP cases may have implored a sense of 'emotional engagement', due to trainees' actions and decisions affecting the outcomes of the VP simulation; which has not been previously identified with less interactive case studies [36,37]. The interactive nature of the VPs in this study and the ability of the software to respond to user input, altering the path through the decision tree to produce different patient outcomes and providing individualised feedback, may have encouraged participants to take responsibility for and reflect on their actions. This can only be beneficial in ensuring deeper learning takes place and preparing individuals to be lifelong learners. Active learning pedagogies have been shown to improve pharmacy student understanding of a range of concepts, greater retention of knowledge and higher satisfaction compared with more passive methods of teaching [38–40]. Learning tools should reflect the level of education that individuals are at and allow a spirality of content in which topics, themes and subjects are integrated and revisited with increasing levels of difficulty, ensuring that new learning is linked to previous learning and the competence of students' increases in-line with the difficulty [41].

Pre-registration trainees commented on their self-confidence after completing the case studies. Previous research has established that pharmacists lack confidence in their own abilities which can prevent them from providing services, counselling or treatment [42,43]. Self-confidence is a key element in the successful integration of knowledge into practice; interactive training tools have been found to increase pharmacists' and undergraduate students' confidence at providing certain services [42,44–47]. A benefit of the VP simulations is their ability to provide a safe environment for users to make mistakes without harming a real patient. The VP technology utilised in this research provided individualised animated spoken and textual feedback to each trainee, which was specific to what they had done well and emphasised areas for improvement. This level of specificity may have encouraged users to repeat the simulations, finding a different way through the decision tree and producing a different outcome with new feedback. Repetition is an essential feature to aid learning in high-fidelity simulations, and the web-based aspect of VPs enables repetitive practice with increased standardisation over traditional real-person interactions [7]. Trainees who used the NI cases provided comments which were more focused on feeling confident for the pre-registration examination, whereas those from the VP group also commented on their confidence for future practice. The VP simulations allowed trainees to understand the benefit of their learning for their role as a pharmacist. Improvements to the VP simulations were suggested and will be acted upon where appropriate, as having a range of accessible resources available for individuals to engage with may be beneficial for their self-directed learning. Previous literature reports the most common problems with VPs as the lack of training and difficulty in using the tools, which have been found to cause user frustration but have not detracted from the learning experience [33,35,47]. Similar findings were reported in this study, with trainees using the technical issues as an opportunity to think about rewording or rephrasing questions.

## Strengths and limitations

The research undertaken for this paper adds considerably to the under-researched area of VPs in pharmacy and especially in pre-registration training. The evaluation of both VP and NI case studies adds to the originality of the work; the majority of previously published literature has evaluated VPs alone, with no alternative learning tool, especially within the area of pharmacy.

This research was not designed in a way to separate the impact of the interactive component and visual avatar; further research is warranted to determine how these elements can impact learning when used separately versus in combination.

Views were gathered from trainees with a mixed demographic profile who had used either one or both of the learning tools and wanted to provide feedback. Although the participants in both groups were mostly female, this is consistent with the distribution profile of pharmacists in practice [48]. Data saturation seemed to occur by interview 15 but as the rest of the interviews were scheduled, they were conducted, which allowed for greater cementing of themes. Conducting the interviews by telephone did not seem a significant barrier to data collection. JT was known to the trainees as a researcher and qualified pharmacist, thus there was the risk that participants may not have felt comfortable voicing their true opinions. Although not all participants spoke freely during the interviews, most did and provided insightful comments which seemed to be honest and included criticisms of the tools or their pre-registration training. Telephone interviews do not easily allow for non-verbal cues to be captured, however reflective notes were made throughout the interviews of participant's comments and non-verbal inferences (i.e. pauses or background noise which may indicate distractions) to ensure contextualisation upon transcription. JT tried to present the participants' perspectives in an open and honest manner, but the analysis was a result of an interaction between the researcher, the pre-registration trainees and a variety of factors (e.g. environment and time of interview) which are recognised as potential influences.

## Conclusions

Pre-registration pharmacists reported that both the VP and NI case studies improved their knowledge, however the VP simulations were also attributed to more complex skill and self-confidence development. Trainees were enthusiastic about VPs as a novel training tool and reportedly valued it as providing a different learning experience than resources currently available; providing trainees a chance to practice realistic scenarios in a safe environment. VPs offer the potential to 'bridge the gap' in pre-registration training variation, provide experiences trainees may otherwise not have, promote learning through reflection on doing and increase overall preparedness for practice.

## Supporting information

**S1 File. Semi-structured interview guide.**
(PDF)

**S1 Table. Topic areas identified for future case studies.** Displays the frequency of topics suggested by trainees which they felt they could have had more support on during the pre-registration year. Each overall topic is subdivided into more specific topics which were reported by trainees. The 'total number of participants' illustrates the overall number of different participants who commented on that topic area.
(PDF)

## Acknowledgments

The authors gratefully acknowledge the time of the pre-registration trainees who participated in the study. Thanks also goes to the pharmacists who helped in the development of the cases, and the digital development team at Keele University.

## Author Contributions

**Conceptualization:** Simon White, Stephen Chapman.

**Data curation:** Jessica Thompson.

**Formal analysis:** Jessica Thompson.

**Investigation:** Jessica Thompson.

**Methodology:** Jessica Thompson, Simon White, Stephen Chapman.

**Project administration:** Jessica Thompson.

**Resources:** Jessica Thompson.

**Software:** Jessica Thompson, Simon White, Stephen Chapman.

**Supervision:** Simon White, Stephen Chapman.

**Validation:** Jessica Thompson, Simon White, Stephen Chapman.

**Visualization:** Jessica Thompson, Simon White, Stephen Chapman.

**Writing – original draft:** Jessica Thompson.

**Writing – review & editing:** Simon White, Stephen Chapman.

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
