## [Decision Letter · Decision Letter 0]

4 Mar 2020

PONE-D-20-02573

Interactive clinical avatars as a tool for training pre-registration pharmacists and increasing their preparedness to practice: a qualitative study.

PLOS ONE

Dear Dr Thompson,

Thank you for submitting your manuscript to PLOS ONE. After careful consideration, we feel that it has merit but does not fully meet PLOS ONE’s publication criteria as it currently stands. Therefore, we invite you to submit a revised version of the manuscript that addresses the points raised during the review process.

We would appreciate receiving your revised manuscript by Apr 18 2020 11:59PM. To enhance the reproducibility of your results, we recommend that if applicable you deposit your laboratory protocols in protocols.io, where a protocol can be assigned its own identifier (DOI) such that it can be cited independently in the future. For instructions see: http://journals.plos.org/plosone/s/submission-guidelines#loc-laboratory-protocols

We look forward to receiving your revised manuscript.

Kind regards,

Vijayaprakash Suppiah, PhD

Academic Editor

PLOS ONE

Journal Requirements:

Journal Requirements:

2. Please include a copy of the interview guide used in the study, in both the original language and English, as Supporting Information, or include a citation if it has been published previously.

Reviewers' comments:

Reviewer's Responses to Questions

**Comments to the Author**

1. Is the manuscript technically sound, and do the data support the conclusions?

Reviewer #1: Partly

Reviewer #2: Yes

2. Has the statistical analysis been performed appropriately and rigorously? 

Reviewer #1: N/A

Reviewer #2: No

3. Have the authors made all data underlying the findings in their manuscript fully available?

Reviewer #1: Yes

Reviewer #2: No

4. Is the manuscript presented in an intelligible fashion and written in standard English?

Reviewer #1: Yes

Reviewer #2: Yes

5. Review Comments to the Author

Reviewer #1: This exploration of online tools for developing educational materials

to pharmacy trainees is potentially interesting and relevant to those

in the field, but not necessarily as portrayed by the authors.

Although the paper does not stand up well as is, a re-framing and

appropriate revision of the discussion and conclusions might make this

paper more defensible and, potentially, more interesting.

The premise of this paper is reasonable to start with. Innovative

approaches to providing training to pharmacy students are certainly

worthy of research and exploration, and interactive avatars seem worth

a try, particularly if they haven't been used before. The

juxtaposition of the avatars and a non-interactive version is a good

step, allowing for the possibility of comparative analysis.

However, confusions in my reading of the methods leave me with

significant concerns regarding the validity of some of the

claims made in the paper. As I understand the methods, this study was

primarily a between-subjects study, with participants using either the

avatar-based or the non-interactive interfaces. My confusion hinges on

the cryptic comment "At the end of the three months, participants who

completed all three cases were provided with the comparative case

studies and invited via email to participate in a telephone

interview."

I do know that this last phrase means, but it seems particularly

important. If each participant only saw one interface, any comparative

comment about liking one interface more than the other (there are

several such comments in the manuscript) would be completely invalid,

as no participant would have any basis for comparison. If this comment

is to be interpreted to mean that participants saw both versions,

there might be some basis for comparison, but as we have no

information about how long each participant spent with the second

interface, we have no way to judge the validity of any

comparisons. Furthermore, we are later told that only three participants

looked at both interfaces, which effectively eliminates any hope of

direct comparison. These methods need to be clarified, and any comparisons

between the two interfaces need to be justified.

Even if these changes are made, I am concerned about inappropriate

quantitative comparisons between the designs. From the abstract:

"Pre-registration trainees appeared overall to prefer learning via the

interactive clinical avatars over the non-interactive case studies and

they were associated with greater self-reported development of

knowledge, skills and confidence for practice." This is a quantitative

claim that should be accompanied by measurements and statistical

tests. Barring that, these claims should be removed.

I have no beef with the qualitative analysis or the thematic

discussion, but I was struck by how minor the differences between the

designs appear to be in the end. This might in itself be noteworthy -

perhaps the realistic content is really what's important here?

I suggest clarifying the methods and avoiding any claims that imply

direct comparison or quantitative analysis.

Other minor notes:

1. The first paragraph in the introduction is a bit of a

mess. Consider focusing the thinking to make a stronger introduction.

2. "Pre-registration" is a very specialized term that might not have

any meaning to readers outside of the field. Suggest removing that

phrase and substituting "pharmacist trainees" or some other

alternative.

3. I have no beef with telephone interviews per se, but in the 2020

world of ubiquitous web conference and screen sharing, citing a 1995

article in support of telephone calls seems a bit anachronistic. Why

not use web conferencing?

4. More details are needed on the construction of the avatar-based

training. This was clearly more than "standard HTML" How were the

avatars built?

5. The 38 topic areas identified in the interviews should be provided

along with the paper. This is potentially useful information.

Reviewer #2: Strengths

* The work is well motivated, well written, and describes a well conducted research

* The authors are familiar with the literature and do not overstate claims (they are aware of work in medicine and nursing that has established similar findings over a decade ago in some cases)

* The findings are valuable to the pharmacology education community

Weaknesses

* The findings already mirror what has been found in other domains some time back so have limited applicability to advancing virtual patient design/research. The results might only be new information to pharmacology education researchers

* Incorrect terminology is used throughout. Please review closely with domain experts.

- Agents vs avatars. what you are using are "agents" - virtual humans controlled by algorithms, not "avatars" virtual humans controlled by humans. avatars and virtual humans are not synonyms.

- Markov model - you can't have a "markov model decision tree", they mean very different things. avoid using terminology that is not well understood.

- animations were encoded as mp4 files - this needs much more clarification as mp4 files are a compression format and how animations are encoded into them is unclear.

* it is unclear if the user selected conversation choices (e.g. EHC case) or had a free text opportunity (child illness case)

* virtual patients are now common and thousands of medical and nursing curricula with many commercial systems being used around the world.

* the discussion about limitation about JT interviewing the participants needs to be rewritten. You can't say that you tried to not be biased and that you believe the answers were candid, because fundamentally you can't know that. rephrase to acknowledge the limitation and highlight improvements for future work.

Minor

* This document could use a good edit for grammar errors.

University's (plural)

punctuation issues througout

line 193 run on

*

Overall

the question here is how valuable is replicating the findings from other disciplines (nursing, medicine) where this understanding of virtual patients discussed in this submission has been known for about a decade is still valuable. the work done here is sound and is well written. however the findings are dated... except for as the authors mentioned the focus on pharmacology training. so if that's still valuable to your readership and academic community (your citations would be primarily from pharmacology educators since others would likely already know the findings in other domains), then this is an accept. if not, then this is a reject since no amount of rephrasing will address this fundamental issue.

6. PLOS authors have the option to publish the peer review history of their article (what does this mean?). If published, this will include your full peer review and any attached files.

Reviewer #1: Yes: Harry Hochheiser

Reviewer #2: No

---

## [Author Response · Author response to Decision Letter 0]

17 Jun 2020

Academic Editor Comments

 Please ensure that your manuscript meets PLOS ONE's style requirements, including those for file naming

The PLOS ONE style templates were reviewed and article/figures/supplementary files amended appropriately. 

Please include a copy of the interview guide used in the study, in both the original language and English, as Supporting Information, or include a citation if it has been published previously.

Interview guide has been included as Supplementary File 1. 

Reviewer #1 Comments

This exploration of online tools for developing educational materials to pharmacy trainees is potentially interesting and relevant to those in the field, but not necessarily as portrayed by the authors. Although the paper does not stand up well as is, a re-framing and appropriate revision of the discussion and conclusions might make this paper more defensible and, potentially, more interesting.

The premise of this paper is reasonable to start with. Innovative approaches to providing training to pharmacy students are certainly worthy of research and exploration, and interactive avatars seem worth a try, particularly if they haven't been used before. The juxtaposition of the avatars and a non-interactive version is a good

step, allowing for the possibility of comparative analysis.

Confusion in my reading of the methods leave me with significant concerns regarding the validity of some of the claims made in the paper. 

• As I understand the methods, this study was primarily a between-subjects study, with participants using either the avatar-based or the non-interactive interfaces. My confusion hinges on the cryptic comment "At the end of the three months, participants who completed all three cases were provided with the comparative case studies and invited via email to participate in a telephone interview." I do know that this last phrase means but it seems particularly important 

The phrase was amended to remove any links to the case studies being comparable. Detail was added to ensure the phrase was clearer regarding the opportunity for pre-registration trainees to use the other style of case study as a learning tool. 

• If each participant only saw one interface, any comparative comment about liking one interface more than the other (there are several such comments in the manuscript) would be completely invalid, as no participant would have any basis for comparison – Comments which relate to comparisons between the two learning tools have been removed throughout the article. An interview guide was created which asked questions on the same topics, however any claims to a comparison of the learning tools have been removed; except where trainees did use both of the tools and provided such comments. However it is made clear that this only relates to a small proportion of the participants. 

• If this comment is to be interpreted to mean that participants saw both versions, there might be some basis for comparison, but as we have no information about how long each participant spent with the second interface, we have no way to judge the validity of any comparisons.

Specific comments relating to comparisons have been removed. Information regarding how long participants had access to the learning tools has been added to the article. Participants were able to access each case study for one month sequentially, to give a total intervention period of three months. 

• Furthermore, we are later told that only three participants looked at both interfaces, which effectively eliminates any hope of direct comparison. These methods need to be clarified, and any comparisons between the two interfaces need to be justified.

It has been clarified that all trainees were given the opportunity to use the other type of learning tool. Only four trainees reported that they had done and provided comments as such. Specific comments relating to comparisons have been removed and made clear when the participants had used both types of learning tool.

• Even if these changes are made, I am concerned about inappropriate quantitative comparisons between the designs. From the abstract: "Pre-registration trainees appeared overall to prefer learning via the interactive clinical avatars over the non-interactive case studies and they were associated with greater self-reported development of knowledge, skills and confidence for practice." This is a quantitative claim that should be accompanied by measurements and statistical

tests. Barring that, these claims should be removed.

Any statements which could be seen as quantitative claims have been rewritten to appropriately report qualitative comments.

• I have no beef with the qualitative analysis or the thematic discussion, but I was struck by how minor the differences between the designs appear to be in the end. This might in itself be noteworthy - perhaps the realistic content is really what's important here?

More information has been added into the manuscript regarding the realism of the virtual patient. Also, more detail which relates to the non-interactive cases being above what trainees would normally have has been added.

• I suggest clarifying the methods and avoiding any claims that imply direct comparison or quantitative analysis.

Amended throughout as noted above.

Other minor notes:

1.The first paragraph in the introduction is a bit of a mess. Consider focusing the thinking to make a stronger introduction.

Paragraph 1 of the introduction has been amended. 

2. "Pre-registration" is a very specialized term that might not have any meaning to readers outside of the field. Suggest removing that phrase and substituting "pharmacist trainees" or some other alternative.

Pre-registration pharmacist training is explained in paragraph 1 of the introduction. It was felt important to use this term to distinguish between undergraduate pharmacist trainees where the majority of previous research has been focused and postgraduate phamacists trainees who may be qualified pharmacist undertaking a further postgraduate course (e.g. prescribing). 

3. I have no beef with telephone interviews per se, but in the 2020 world of ubiquitous web conference and screen sharing, citing a 1995 article in support of telephone calls seems a bit anachronistic. Why not use web conferencing?

A more up-to-date reference has been added to the article. Telephone interviews were used as they were identified as the least unobtrusive method. Some of the interviews took place during participants’ working hours and they could easily use a phone. However, more technologically advanced methods would be considered in future research.

4. More details are needed on the construction of the avatar-based training. This was clearly more than "standard HTML" How were the avatars built?

More information has been added into the article. The page layout was defined using HTML with CSS to define the position and colour of screen elements. For the VP cases using free-text variations, once the 'speak to patient' button was clicked the text was sent to a web service hosted on the same server to process the text. This then returned a code to the site which was processed by the client side Javascript (using the jQuery library) to determine what animation to play and how the case should behave next. The MCQ cases processed everything client side, skipping the web service step.

5. The 38 topic areas identified in the interviews should be provided along with the paper. This is potentially useful information.

A table showing the topic areas has been included as Supplementary File 2.

Reviewer #2: 

Strengths

* The work is well motivated, well written, and describes a well conducted research

* The authors are familiar with the literature and do not overstate claims (they are aware of work in medicine and nursing that has established similar findings over a decade ago in some cases)

* The findings are valuable to the pharmacology education community

Weaknesses

• The findings already mirror what has been found in other domains some time lack so have limited applicability to advancing virtual patient design/research. The results might only be new information to pharmacology education researchers.

It is agreed that the findings are similar to previous research in other healthcare professions. The article has tried to infer the greater advancement this brings to the pharmacy profession and has focused on the use of free-text virtual patients which are under-researched in the literature. 

• Incorrect terminology is used throughout. Please review closely with domain experts:

o Agents vs avatars. what you are using are "agents" - virtual humans controlled by algorithms, not "avatars" virtual humans controlled by humans. avatars and virtual humans are not synonyms 

Terminology changed to ‘virtual patients’ throughout. 

o Markov model - you can't have a "markov model decision tree", they mean very different things. avoid using terminology that is not well understood.

Markov model removed. 

• Animations were encoded as mp4 files - this needs much more clarification as mp4 files are a compression format and how animations are encoded into them is unclear.

More information has been added into the article. Characters were modelled, textured and animated using Autodesk's Maya 3D package. Once animated, a series of still images were rendered using the rendering package Mental Ray supplied with Maya. The still images were then composited using Adobe After Effects to create the final .mp4 animation files.

• It is unclear if the user selected conversation choices (e.g. EHC case) or had a free text opportunity (child illness case)

How the user interacted with each virtual patient simulation has been added into the methods section and it has been made clear which utilised multiple-choice vs free-text input. 

• Virtual patients are now common and thousands of medical and nursing curricula with many commercial systems being used around the world

This is true. The differences of this virtual patient system (free-text, animated feedback) has been added into the manuscript. Implications for healthcare training have been made clearer. 

• The discussion about limitation about JT interviewing the participants needs to be rewritten. You can't say that you tried to not be biased and that you believe the answers were candid, because fundamentally you can't know that. Rephrase to acknowledge the limitation and highlight improvements for future work.

The limitation regarding JT interviewing the participants has been rewritten. 

Minor

• This document could use a good edit for grammar errors

A thorough proofread has been done and edited where appropriate.

---

## [Decision Letter · Decision Letter 1]

29 Jun 2020

PONE-D-20-02573R1

Virtual Patients as a tool for training pre-registration pharmacists and increasing their preparedness to practice: a qualitative study.

PLOS ONE

Dear Dr. Thompson,

Thank you for submitting your manuscript to PLOS ONE. After careful consideration, we feel that it has merit but does not fully meet PLOS ONE’s publication criteria as it currently stands. Therefore, we invite you to submit a revised version of the manuscript that addresses the points raised during the review process.

Dear Authors,

Please address the comments by the first reviewer. 

Thank you.

We look forward to receiving your revised manuscript.

Kind regards,

Vijayaprakash Suppiah, PhD

Academic Editor

PLOS ONE

Reviewers' comments:

Reviewer's Responses to Questions

**Comments to the Author**

1. If the authors have adequately addressed your comments raised in a previous round of review and you feel that this manuscript is now acceptable for publication, you may indicate that here to bypass the “Comments to the Author” section, enter your conflict of interest statement in the “Confidential to Editor” section, and submit your "Accept" recommendation.

Reviewer #1: (No Response)

Reviewer #2: All comments have been addressed

2. Is the manuscript technically sound, and do the data support the conclusions?

Reviewer #1: Yes

Reviewer #2: Yes

3. Has the statistical analysis been performed appropriately and rigorously? 

Reviewer #1: N/A

Reviewer #2: Yes

4. Have the authors made all data underlying the findings in their manuscript fully available?

Reviewer #1: No

Reviewer #2: Yes

5. Is the manuscript presented in an intelligible fashion and written in standard English?

Reviewer #1: Yes

Reviewer #2: Yes

6. Review Comments to the Author

Reviewer #1: Thanks for your thorough responses to initial comments. I have one substantive comment and a few remaining comments that I would like to see addressed:

1. My main conceptual concern is that your study essentially introduces two interlinked interventions: 1. The use of the conversational agent, and 2) the use of the visual avatars. It is possible that both of these are needed to have a strong impact on educational experiences - this study is not designed in a way that would answer that question. However, I would like to see something in the discussion about the possibility that the interactive component on its own might or might not be as useful as the virtual patients described in this paper.

2. The term "pre-registration" is still used in the abstract in a manner that I found confusing. If this were a journal that was (a) focused on pharmacy education, and (b) focused on a British audience, this usage would be acceptable, but as neither of these things are true, this language obscures rather than clarifies. I suggest revising to "pre-certification", internship or something similar.

3. The abstract refers to the "framework approach". I assume this is referring to the five-stage framework approach used for qualitative analysis? This should be clarified.

4. The abstract makes this significant claim: "however those who used the virtual patient commented on the development of real-life complex skills and aspects of learning, and those who used the non-interactive cases focused on more simple skills and knowledge." I don't see this justified in the presentation of the results or the discussion. I suggest reframing.

4. under "Participants", please clarify what is meant by "sector of training".

5. Line 160-161. I presume MCQ means "multiple-choice question" , but it is not defined, and there is no indication of which of the scenarios involved this type of question.

6. Results. 20 interviews were conducted, but earlier we were told that 56 participants were eligible. Did 39 decline? this should be clarified.

7. The gender differential (heavily skewed toward female in both cases) should be mentioned in the discussion as having a possible impact on the observations.

Reviewer #2: The authors did a reasonable job addressing issues raised and the clarity of this submission is much improved.

7. PLOS authors have the option to publish the peer review history of their article (what does this mean?). If published, this will include your full peer review and any attached files.

Reviewer #1: **Yes: **Harry Hochheiser

Reviewer #2: No

---

## [Author Response · Author response to Decision Letter 1]

6 Aug 2020

Thank you for your review of my manuscript. I have addressed your comments below. 

1. My main conceptual concern is that your study essentially introduces two interlinked interventions: 1. The use of the conversational agent, and 2) the use of the visual avatars. It is possible that both of these are needed to have a strong impact on educational experiences - this study is not designed in a way that would answer that question. However, I would like to see something in the discussion about the possibility that the interactive component on its own might or might not be as useful as the virtual patients described in this paper.

- This point is acknowledged and agree that the research was not designed in a way to separate the impact of the conversation agent and visual avatar. The manuscript has been amended (line 485-488) “This research was not designed in a way to separate the impact of the interactive component and visual avatar; this warrants further research to determine how these elements can affect learning when used separately versus in combination.”

2. The term "pre-registration" is still used in the abstract in a manner that I found confusing. If this were a journal that was (a) focused on pharmacy education, and (b) focused on a British audience, this usage would be acceptable, but as neither of these things are true, this language obscures rather than clarifies. I suggest revising to "pre-certification", internship or something similar.

- The abstract has been amended to clarify what pre-registration training means. Line 20-22 “In the United Kingdom, pre-registration training refers to a year of workplace based training which pharmacy graduates must complete prior to professional registration as pharmacists.”

3. The abstract refers to the "framework approach". I assume this is referring to the five-stage framework approach used for qualitative analysis? This should be clarified.

- This has been clarified in the abstract – line 28.

4. The abstract makes this significant claim: "however those who used the virtual patient commented on the development of real-life complex skills and aspects of learning, and those who used the non-interactive cases focused on more simple skills and knowledge." I don't see this justified in the presentation of the results or the discussion. I suggest reframing.

- The abstract has been amended (lines 34-37) “those who used the virtual patient commented on the development of communication skills and an increase in confidence for practice and those who used the non-interactive cases focused on knowledge acquisition and numeracy”. The discussion has also been amended to reflect the changes (lines 407-409).

5. Under "Participants", please clarify what is meant by "sector of training".

- This has been clarified on line 107 (community or hospital pharmacy).

6. Line 160-161. I presume MCQ means "multiple-choice question" , but it is not defined, and there is no indication of which of the scenarios involved this type of question.

- Line 162-163 has been amended to ‘multiple–choice input cases’. The explanation of which cases used which input style was provided previously: lines 145-146 “The EHC simulation utilised both multiple-choice input and free-text input”, line 150 “They were able to interact with the VP via multiple-choice input”, line 151 “The childhood illness VP simulation utilised free-text input”.

7. Results. 20 interviews were conducted, but earlier we were told that 56 participants were eligible. Did 39 decline? this should be clarified.

- The manuscript has been amended (line 214-215) “The remaining 36 participants who were invited for a telephone interview did not respond to the invitation.”

8. The gender differential (heavily skewed toward female in both cases) should be mentioned in the discussion as having a possible impact on the observations.

- The manuscript has been amended (lines 491-493) “Although the participants in both groups were mostly female and could have impacted the data obtained, this is consistent with the distribution profile of pharmacists in practice [48].”

---

## [Editor Report · Decision Letter 2]

13 Aug 2020

Virtual Patients as a tool for training pre-registration pharmacists and increasing their preparedness to practice: a qualitative study.

PONE-D-20-02573R2

Dear Dr. Thompson,

We’re pleased to inform you that your manuscript has been judged scientifically suitable for publication and will be formally accepted for publication once it meets all outstanding technical requirements.

Kind regards,

Vijayaprakash Suppiah, PhD

Academic Editor

PLOS ONE

---

## [Editor Report · Acceptance letter]

21 Aug 2020

PONE-D-20-02573R2 

Virtual Patients as a tool for training pre-registration pharmacists and increasing their preparedness to practice: a qualitative study. 

Dear Dr. Thompson:

I'm pleased to inform you that your manuscript has been deemed suitable for publication in PLOS ONE. Congratulations! Your manuscript is now with our production department. 

Kind regards, 

on behalf of

Dr. Vijayaprakash Suppiah 

Academic Editor

PLOS ONE